# Volatilome Analyses and In Vitro Antimicrobial Activity of the Essential Oils from Five South African *Helichrysum* Species

**DOI:** 10.3390/molecules25143196

**Published:** 2020-07-13

**Authors:** Basma Najar, Valeria Nardi, Claudio Cervelli, Giulia Mecacci, Francesca Mancianti, Valentina Virginia Ebani, Simona Nardoni, Luisa Pistelli

**Affiliations:** 1Dipartimento di Farmacia, Università di Pisa, Via Bonanno 6, 56126 Pisa, Italy; nardivaleria12@gmail.com (V.N.); g.mecacci@studenti.unipi.it (G.M.); luisa.pistelli@unipi.it (L.P.); 2CREA-Centro di Ricerca Orticoltura e Florovivaismo, Corso Inglesi 508, 18038 Sanremo, Italy; claudio.cervelli@crea.gov.it; 3Dipartimento di Scienze Veterinarie, Università di Pisa, Viale delle Piagge 2, 56124 Pisa, Italy; francesca.mancianti@unipi.it (F.M.); valentina.virginia.ebani@unipi.it (V.V.E.); simona.nardoni@unipi.it (S.N.); 4Centro Interdipartimentale di Ricerca Nutraceutica e Alimentazione per la Salute “Nutrafood”, Università di Pisa, Via del Borgetto 80, 56124 Pisa, Italy

**Keywords:** *H. cooperi*, *H. edwardsii*, *H. pandurifolium*, *H. odoratissimum*, *H. trilineatum*, HS-SPME, GC-MS, Kirby-Bauer agar disc diffusion, microdilution test

## Abstract

*Helichrysum* genus was used in folk South African medicine to treat various human disorders. As a part of our on-going research addressing the exploitation of South African plants belonging to this genus, five species were investigated for their volatile and antimicrobial activities. The volatile organic compounds (VOCs) and the essential oils (EOs) were analysed by gas chromatography mass spectrometry (GC-MS). Microdilution was the method used for assessing both antimycotic and antibacterial activities, which was also tested by Kirby-Bauer agar disc diffusion. Total monoterpenes (TMs) dominated the VOCs of four species (*H. trilineatum* (70.6%), *H. edwardsii* (79.3%), *H. cooperi* (84.5%), and *H. pandurifolium* (57.0%)). *H. cooperi* and *H. edwardsii* EOs showed the predominance of TMs (68.2% and 84.5%, respectively), while *H. pandurifolium* and *H. trilineatum* EOs were characterized by the prevalence of TSs (86.5% and 43.6%, respectively). *H. odoratissimum* EO evidenced a similar amount of both TMs (49.5%) and TSs (46.4%). *Microsporum canis* was more sensitive to these EOs. The lowest minimum inhibitory concentration (MIC) was observed with *H. pandurifolium* and *H. edwardsii* EOs (0.25%). *H. pandurifolium* and *H. trilineatum* had a good effect on *Staphylococcus aureus* (MIC 5%). These findings open new perspectives for the exploitation of these natural compounds for application in cosmetics and pharmaceutics.

## 1. Introduction

Among the 600 species listed in the *Helichrysum* genus (Asteraceae family), almost forty percent occur in South Africa. They are subdivided into 30 morphological groups where the only characters that differ are the shape and size of the flower heads [1]. Local indigenous people widely used these plants since antiquity and then enticed the interest of scientists by their richness in unusual secondary metabolites. Numerous organic solvents were used to extract these metabolites and to benefit from their biological activity.

In 2008, Lourens [2] with colleagues published a review about the traditional uses, biological activity, and phytochemistry of South African *Helichrysum* species and enumerated seventy species. Three years later, the same team investigated the antimicrobial effects and in vitro cytotoxicity of these same species together with six new ones [3]. Since then, few new species were investigated even though new reports were present in the literature to consolidate and confirm what was previously found in the species and already cited; Lourens et al. Mamabolo [4], for example, have evaluated the antigonorrhea and cytotoxicity of *H. caespititium* (DC.) Harv. This species together with *H. odoratissumum* L. Sweet and *H. petiolare* Hilliard and B.L. Burtt were the subject of three reviews published recently by Maroyi [5,6,7], who brought to days the traditional medicinal uses, phytochemistry, and biological activities of the cited species.

Besides the species already mentioned by Lourens, new taxa were studied such as *H. niveum* Graham reported by Popoola [8]. Yadizi, in his turn, studied the anti-HIV screening of extracts from thirty-two South African *Helichrysum* spp., some of them (nineteen) never reported before [9]. However, the authors cited hitherto focused on extracts with organic solvents of the South African *Helichrysum* species rather than on the essential oils, obtained by hydrodistillation or steam distillation. Turning back to the Lourens reports, before his first review, in 2004, he published an article on the biological activity of the essential oils of four indigenous species [10]. It did not prevent other researchers, fascinated by the biological activities of *Helichrysum* essential oils, from also investing in several species such as *H. kraussii* Sch. Bip together with *H. rugulosum* Less [11], *H. aureonitens* Sch. Bip. essential oil (EO) [12], and *H. cymosum* (L.) D. Don [13]. *H. odoratissimum* was analysed by Asekun, who investigated the effect of drying methods on the essential oil compositions dete [14]. In the last decade, essential oils of other species were also considered such as *H. foetidum*, investigated by Samie and co-workers [15], who focused on the antimicrobial activity of the EO of this species without revealing its chemical composition. Moreover, the EOs of the species *H. cymosum*, *H. petiolare* [16], *H. odoratissimum* [16,17], and *H. splendidum* [18,19] were also investigated. All these authors used plant material collected directly from South Africa; others, on the contrary, decided to cultivate seeds imported from this region in different environments. As a part of our on-going collaboration with CREA (Centro di Ricerca Orticoltura e Florovivaismo) in Sanremo (Italy), we focused on the exploitation of South African *Helichrysum* species for ornamental and industrial application, and *H. nudifolim* [20], *H. cymosum* L., and *H. tenax* M.D. Hend [21], as well as other eight species [22], were studied for their volatilome.

Continuing in this project, five new South African species of *Helichrysum*, grown in Italy, were the subject of this work where, after the first step on the chemical composition of both spontaneous emissions and essential oils, the antimycotic and antibacterial activities of the essential oils were assessed.

## 2. Results and Discussion

### 2.1. Aroma Profile

The aromatic profile of the studied species of *Helichrysum* reveals the presence of 32 different compounds with a percentage of identification ranging from 99.8% to 100% (Table 1). Only four compounds are common in all the species, even though with different percentages. These constituents were sabinene (0.1% in *H. pandurifolium* to 34.4% in *H. cooperi*), *β*-pinene (from 2.5% in *H. cooperi* to 17.5% in *H. pandurifolium*), *α*-copaene (from 0.2% in *H. trilineatum* to 2.5% of *H. cooperi*), and *β*-caryophyllene (0.7% in *H. cooperi* and 6.9% in *H. edwardsii*).

All the spontaneous emissions of the studied species were characterized by a high percentage of monoterpene hydrocarbons (Figure 1, Table 2). The amount ranged from 54.7% in *H. pandurifolium* to more than 69% of the total identified fraction in *H. trilineatum*. Sabinene (34.4%) and 1,8-cineole (20.5%) were the predominant constituents in *H. cooperi*, while *α*-pinene was the major one in *H. pandurifolium* and *H. trilineatum* (25.7% and 64.8%, respectively) followed by a good percentage of *β*-pinene (17.5%) in the case of *H. pandurifolium*. However, more than the 80% of monoterpene compounds in *H. edwardsii* were represented by sabinene (29.2%), *β*-pinene (16.4%), and *β*-thujone (18.0%). Sesquiterpene hydrocarbons was the second main class of compounds and the highest amount was evidenced in *H. pandurifolium* (35.2%). *δ*-cadinene was of the main component, which represented more than 52% of the total sesquiterpenes.

Principal component analysis (PCA) as well as hierarchical cluster analysis (HCA) (Figure 2 and Figure 3) were performed for all compounds present in at least one of the investigated species with a percentage higher than 2%. Taking into consideration the PC1 axis (the direction explaining the maximum variance (71.5%)), two macro groups were present: one with positive loading along this axis and the other with a negative one. It is important to notice that all the variables are concentrated around the origin of the axes, and only four variables, *δ*-cadinene, *α*-pinene, sabinene, and *β*-thujone, were more dispersed. In particular, these compounds explained the position of each studied *Helichrysum* in the PCA graph. In fact, the positive loading of *δ*-cadinene in both axes was responsible for the *H. pandurifolium* VOC position. *H. trilineatum* located in the lower case of the graph (negative PC1, positive PC2) owing its position to the high amount in *α*-pinene. The highest amount of sabinene placed *H. cooperi* in the left down quadrant, while *H. edwardsii* was found in the upper case on the left owing to its amount in *β*-thujone.

Then, the HCA of the volatile emissions from each *Helichrysum* species, shown in Figure 2, pointed out an outcome perfectly in agreement with the PCA results. HCA clustered these *Helichrysum* species in two different groups: A and B, where *H. pandurifolium* and *H. trilineatum* made up group A, while group B clustered together the remaining species.

Only few studies are reported in the literature on the aromatic profile of the *Helichrysum* species from South Africa. The work of Reidel [20] pointed out a high percentage of sesquiterpene hydrocarbons (92.9%) in the flowers of *H. nudifolium* (L.) Less., with *β*-caryophyllene as the most abundant constituent (79.4%). In the present work, the studied species showed a different behaviour; in fact, the dominant class of compounds was monoterpene hydrocarbons. This comportment did not deny the presence of sesquiterpene compounds especially in *H. pandurifolium*. *β*-caryophyllene was also present in all the studied species, but with a lesser amount. 

Focusing our attention on the South African species of *Helichrysum*, recently, Giovanelli [21] and Najar [22] investigated the VOC emissions of the following species: *H. cymosum* (L.) D.Don, *H. odoratissimum* (L.) Sweet, *H. petiolare* Hilliard & B.L.Burtt and *H. tenax* M.D.Hend., *H.foetidum* (L.) Cass., *H. incarnatum* DC., *H. indicum* (L.) Grierson, *H. montanum* DC., *H. mutabile* Hilliard, *H. patulum* (L.) D. Don, and *H. setosum* Harv. More than 45% of the total volatile composition from all the listed species was done by monoterpene hydrocarbons, with *α*- and *β*-pinene (in *H. odoratissimum*, *H. petiolare*, *H. tenax*, *H. patulum* and *H. setosum*), (*Z*)-*β*-ocimene (in *H. cymosum* and *H. incarnatum*), sabinene (*H. foetidum*), myrcene (*H. indicum*), triciclene (*H. montanum*), and limonene (*H. mutabile*) as the main compounds. The investigated species herein showed a similar behaviour concerning the predominant class of constituents. Except for tricyclene, all the other compounds previously mentioned had been identified, even though with different percentages. *H. cooperi* such as the already studied *H. tenax* underlined a high percentage of oxygenated monoterpenes, with the most abundant compound 1,8-cineole [21].

### 2.2. Essential Oils

The chemical composition of the EOs from the analysed species is reported in Table 3. More than eighty compounds were detected, whose percentages accounted for more than 97% of the total identified fraction. The highest yield of the extracted oils was found in *H cooperi* (0.6% *w*/*w*), while *H. trilineatum* showed the lowest one (0.1 *w*/*w*). Only three compounds were in common in all the studied species: *α*-pinene (whose percentage ranged from 2.3% in *H. cooperi* to 16.5% in *H. odoratissimum*), *β*-pinene (from 1.5% in *H. trilineatum* to 31.2% in *H. edwardsii*), and *β*-caryophyllene (from 0.4% in *H. trilineatum* to 4.1% in *H. odoratissimum*).

*H. pandurifolium*, less rich in the number of identified compounds (24), was dominated by the oxygenated sesquiterpenes (os, 74.7%) (Figure 4, Table 4) mostly represented by viridiflorol (60.3%). This EO pointed out almost the same amount in sesquiterpene and monoterpene hydrocarbons (11.8% and 10.3%, respectively) (Figure 4), and showed five exclusive compounds: pogostol (4.8%), germacrene d-4-ol (2.5%), (*Z*)-*β*-ocimene (1.1%), (*E*)-1(6,10-dimethylundec-5-en-2-yl)-4-methylbenzene (1.4%), and aristolochene (0.5%). *H. odoratissimun* EO evidenced a very close percentage of os, om, and mh (28.9%, 25.8%, and 23.7%, respectively) with *epi*-cubebol (9.0%), 1,8-cineol (25.1%), and *α*-pinene (16.5%) as the most representative compound in each class, respectively. It is fair to report that the exclusive constituents represented 19% of the identified fraction, where *epi*-cubebol and 14-hydroxy-9-*epi*-(*E*)-caryophyllene (5.0%) showed the predominant amount. More than 65% of the identified fraction of the EO composition of *H. edwardsii* was represented by monoterpene hydrocarbons (mh) characterized by sabinene (22.4%) and *β*-pinene (31.2%). This latter class was also the main one in *H. cooperi* (mh 37.1%), where sabinene was also present in a good percentage (14.7%). Oxygenated monoterpene (om 31.1%) was also in a good amount in this same EO, especially represented by 1,8-cineol (16.4%).

The exclusivity in a high amount of both *trans* and *cis* thujone (8.7% and 1.3%, respectively) in *H. edwardsii* is noteworthy.

*H. trilineatum* EO accounts for the greatest number of constituents (43) (Table 3), showing a composition dominated by oxygenated diterpenes (od, 28.2%), with the main compound being sandaracopimarinol (17.7%), a peculiar compound of this species. Followed by a high percentage of sesquiterpene hydrocarbons (24.1%) characterized by bicyclogermacrene (10.7%) and germacrene D (5.5%), together with oxygenated sesquiterpenes (19.5%), with spatulenol (7.2%) as the most abundant one. Monoterpene hydrocarbons, which represented 13.2% of the total identified compounds, were also present with *α*-pinene (11.5%) as a major constituent. Abieta-7,13-diene (6.1 %) was indeed the most important compound of the diterpene hydrocarbons, which represented 9.5% of the total composition.

PCA and HCA statistical analyses were performed using the compounds present in amounts greater than 5% for at least one of the studied species (Figure 5 and Figure 6). The upper right quadrant of PCA analysis (Figure 5) included exclusively *H. pandurifolium*, whose main component was viridiflorol. *H. edwardsii* was located in the upper left quadrant, because of their higher percentages in both *β*-pinene and sabinene, compounds lading negatively with PC2. All the others species were in the lower left quadrant (PC 1 negative and PC 2 negative).

The dendrogram obtained from HCA, reported in Figure 6, was in accordance with the results of PCA and distinguished *H. pandurifolium* from all the other *Helichrysum* species located in group A. The group B consists of two subgroups (B1 and B2). B1 included *H. trilineatum* with *H. odoratissimum,* while subgroup B2 clustered together the remaining species.

Although numerous reports in the literature deal with the chemical composition of the EOs from different species of *Helichrysum*.

Regarding the EO composition of *H. odoratissimum*, the first studies date back to 1993 when both Gundidza and Zwaving [23] and Lwande [24] reported *α*-pinene and *α*-humulene as main constituents. Later on, Kuiate and coworkers [25] found EO rich in *α*-pinene (41.0–47.0%), *β*-caryophyllene (5.0–14.0%), and *α*-curcumene (4.0–20.0%), while limonene (23.0–32.0%), *β*-caryophyllene (12.0–13.0%), and *α*-pinene (8.0–10.0%) characterized the EO analysed by Asekun [14]. Except for *α*-curcumene and *α*-humulene, all the other compounds were present in a good amount in the studied *H. odoratissimum* EO. A more recent paper on the EO of the same species [17] pointed out *β*-pinene (51.6%), limonene (16.9%), *α*-humulene (5.6%), and *β*-caryophyllene (4.7%) as major constituents. Limonene was completely absent in the EO studied herein against a high amount in 1,8-cineole. Concerning b-caryophyllene, our result was in agreement with what was found by these latter cited authors.

Cavalli [26] characterized the EOs of six Madagascarian species of *Helichrysum*: *H. gymnocephalum* (DC.) Humbert, *H. bracteiferum* (DC.) Humbert, *H. selaginifolium* (DC.) Viguier & Humbert, *H. cordifolium* DC., *H. faradifani* Scott-Elliot, and *H. hypnoides* (DC.) Viguier & Humbert. The main compounds evidenced were 1,8-cineole (59.7% in *H. gymnocephalum* and 27.3% in *H. bracteiferum*), *β*-pinene (38.2% in *H. selaginifolium*), and *β*-caryophyllene (55.6% in *H. cordifolium*, 34.6% in *H. faradifani*, and 34.0% in *H. hypnoides*).

The EO composition of South African *H. dasyanthum* (Willd.) Sweet, *H. excisum* (Thunb.) Less., *H. petiolare* Hilliard & B.L.Burtt, and *H. felinum* Less. was analysed by Lourens [10]. In the first three species, 1,8-cineol (20.0–34.0%), *α*-pinene (3.0–17.0%), and *p*-cymene predominated (6.0–10.0%), while *β*-caryophyllene (27.6%), *α*-humulene (9.4%), and caryophyllene oxide (6.9%) were the most abundant in *H. felinum* EO. Viridiflorol was present in a good amount (18.2%) only in *H. excisum*.

Previous investigations on other *Helichrysum* species from South Africa were done by our research group and, among these, Bandeira Reidel evidenced *β*-caryophyllene as the most abundant compound (46.0%) in *H. nudifolium* (L.) Less. [20]. More recently, Najar studied other eight species [22]. Except for *H. incarnatum* where monoterpene hydrocarbons prevailed (60.7%), mostly represented by (*Z*)-*β*-ocimene (17.0%), *β*-phellandrene (12%), and *α*-pinene (11.6%), and all the other species highlighted a high percentage of sesquiterpene compounds. Valerianol (in *H. basalticum* Hilliard, *H. foetidum* (L.) Cass. and *H. setosum* Harv), viridiflorol (*H. foetidum* (L.) Cass. and *H. montanum* DC), *β*-caryophyllene (*H. indicum* (L.) Grierson and *H. patulum* (L.) D.Don), sandaracopimarinol (in *H. mutabile* Hilliard), *β*-himachelene, caryophyllene oxide (*H. montanum* DC.,), *δ*-cadinene (*H. montanum* DC), aromadendrene, and globulol (*H. basalticum* Hilliard) were the main constituents. Among all the compounds identified in the previous South African species, only four of them showed more than 10% of the identified fraction of the analysed species in this work. These constituents were 1,8-cineole (in *H. cooperi* and *H. odoratissimum)*, *α*-pinene (in *H. odoratissimum* and *H. trilineatum*), viridiflorol (*H. pandorifolum*), and sandaracopimarinol (*H. trilineatum*).

*β*-caryophyllene seemed to be one of the peculiar compounds in the EO of *Helichrysum* spp. This constituent was also present, in variable percentages, in all our EOs.

### 2.3. Antimicrobial Activity

The antimicrobial analyses were performed in all the studied species, together with two further species (*H. foetidum* and *H. patulum*) whose composition was reported in our previous study [22]. The results of the tests are reported in Table 5 and Table 6.

Dermatophytes were more sensitive to some EOs in comparison with environmental fungi, where no inhibition of their growth was noticed. *M. canis* was the most sensitive fungal species especially to *H. pandurifolium, H. trilineatum*, and *H. edwardsii* EOs, with the lowest values of minimum inhibitory concentration (MIC) (0.25%). These oils were active on *T. mentagrophytes* also (MIC value of 0.5% and 1% for the latter, respectively). Moreover, *H. cooperi* showed a not negligible action on *M. canis* (MIC 0.5%). It was not possible to set an MIC value for *H. patulum* EO owing to its very low yield. The amount obtained allowed us to test its antifungal activity only with an initial dilution of 5%, to which both dermatophytes have been shown to be sensitive. All dermatophytes were sensitive to itraconazole (8 µg/mL), while other fungi scored sensitive to amphotericin B (range: 0.015–8 mg/L).

Both Kirby-Bauer and microdilution methods highlighted that *Staphylococcus aureus* and *Staphylococcus pseudointermedius* were mostly sensitive to *H. pandurifolium* and *H. trilineatum* EOs.

*H. edwardsii* EO produced a growth inhibition zone of 7 mm diameter and an MIC value of 10% against *S. pseudointermedius,* but it was not active versus *S. aureus*. *H cooperi* and *H odoratissimum* did not show anti staphylococcal activity. Both *S. aureus* and *S. pseudointermedius* were sensitive to chloramphenicol (growth inhibition zone of 21 mm in disk diffusion, 8 µg/mL in microdilution method).

The antimicrobial activity of the EOs extracted from endemic plants that grow spontaneously in South Africa was previously investigated [10,11,12,13,18,19]. To the best of our knowledge, the antimicrobial activity of the species studied herein is reported for the first time, except for *H. foetidum* and *H. odoratissimum*. The EO of this latter species grown in South Africa was tested by Lawal [17] and he emphasized the powerful antibacterial action on *S. aureus* (ATCC 3983) and *S. aureus* (ATCC 6538), with MIC of 1.3 and 2.5 mg/mL, respectively. Not much convergence between our results and those previously reported is noted, which could be owing to the differences in the EO compositions. The studied species was rich in 1,8-cineole rather than limonene (case of Lawal), which was previously reported to be less active [27,28,29].

The biological activity of *H. foetidum* was studied by Samie [15,16,17,18,19,20,21,22,23,24,25,26,27,28,29,30,31] and reported in the review of Maroyi [32]. Although the authors did not investigate the EO composition, they described its antifungal and antibacterial activities (MIC value on *S. aureus* >7.5mg/mL). This result was in agreement with our assay.

*H. trilineatum* exhibited a moderate action on both bacterial strains and fungi studied herein and, even though it had an equal amount of spathulenol such as in Tanzanian *H. fulgidum* [33], endowed significant antibacterial activities, and a good percentage of both bicyclogermacrene and germacrene D; likewise in *H. splendidum* collected in Mpumalanga (South Africa) rich in the cited constituents (range of 7.41–20.5% and 11.1–26.5%, respectively), which evidenced a good antifungal activity [18]. Actually, the antibacterial action of this EO could be owing to the presence of sandaracopimarionol, as the purified compound showed a strong antibacterial activity [34].

*H. rugulosum*, widely collected in South Africa, was investigated for its EO composition and biological activity [11]. The authors ascribed the antibacterial properties of its EO to the high percentage of caryophyllene oxide (8.8%) and viridiflorol (3.7%), with an MIC value on *S. aureus* of 0.38 mg/mL. Other plant species rich in this latter compound evidenced a potent antibacterial activity on *S. aureus* such as *Eugenia umbelliflora* (viridiflorol, 17.7%) with MIC = 0.119 mg/mL [35] and *Salvia officinalis* (viridiflorol, 10.93%) (MIC = 0.2 mg/mL) [36]. Fungi were more sensitive to the *H. pandurifolim* EO rather than bacteria. This oil was mostly composed by viridiflorol (more than the half of identified fraction). As no study investigated the antifungal effect of this latter compound, the activity shown by the use of *H. pandurifolim* EO might owing to the synergic effect of the whole constituents, even those present in a lesser amount. In fact, several compounds associated with several EOs demonstrating good antifungal activities are reported in the literature such as *β*-pinene, spathulenol, *δ*-cadinene, and epi-*α*-muurolol [37,38,39,40]. The sensitivity of fungi rather than bacteria was registered again using the *H. cooperi* EO. This activity could be owing to the presence of a good amount of terpinen-4-ol [41] and himachalol [42]. *H. edwardsii* EO had a varying degree of effectiveness, with *Aspergillus* and *Fusarium* being relatively more resistant when compared with dermatophytes. This result was confirmed by the work of Abu-Darwish [43]. The richness in thujone of this oil could explain its effectiveness against dermatophytes, even though several works reported it was active on both bacteria and fungi [44,45].

## 3. Materials and Methods

### 3.1. Plant Materials

All the studied plants, originating from South Africa (Table 7), belong to the collection of the Research units for floriculture and ornamental species of aromatic plants (CREA) located in Sanremo, Italy. The seeds were purchased from specialized companies in sailing seeds of African plant species (Silver Hill-PO Box 53108, Kenilworth, 7745 Cape Town, South Africa and B & T World Seeds-Paguignan, 34210 Aigues Vives, France). The plants were grown in pots under the same edaphic substrate (perlite (2:1 *v*/*v* added with 4 g/L slow release fertilizer) and climatic conditions (Csa in Köppen-Geiger climate classification with an average annual temperature of 16 °C and an annual rainfall of about 700 mm; frosts are light and very rare). After clonal propagation, the plants grew in pots in the open air and were periodically watered. Flowering took place after one year. A voucher sample of each plant was deposited at the herbarium of the Hanbury Botanical Gardens (La Mortola-Ventimiglia, Imperia, Italy). The correct identification of the plants was performed by one of the authors, Claudio Cervelli.

The aerial parts were collected during the flowering period and were dried at room temperature for 5 days up to constant weight. *H. cooperi*, *H. edwardsii*, and *H. odoratissimum* were collected in 2018, while *H. pandurifolium* and *H. trilineatum* were collected in 2019.

### 3.2. Chemical Investigation

#### 3.2.1. Spontaneous Emission Analysis and EO Extraction

Living fresh plant material (1 g) of each species, maintained in pots, was sent from Sanremo (CREA) to Pisa (Dipartimento di Farmacia, University of Pisa, Pisa, Italy) and was maintained in a greenhouse for 2–3 days before SPME analysis. The HS-SPME (head space-solid phase microextraction) analyses were performed using 100 μm polydimethylsiloxanes (PDMS) fibre manufactured by Supelco Ltd. (Bellefonte, PA, USA). Prior to the analyses, the fibre was conditioned according to the manufacturer’s instruction, at 250 °C for 30 min in the injector of a gas chromatograph. The refined material, with a weight of 3.00 ± 0.01 g, was placed in a 50 mL glass vial and sealed with an aluminum cap for 60 min (equilibration time). Exposition of the fibre in the headspace phase of the samples took place for 15 to 30 min at a temperature of 23 °C. Subsequently, the fibre was transferred to the injector of the gas chromatograph (temperature 250 °C), where the analytes were thermally desorbed [47]. The composition of the compounds desorbed from SPME fibre was examined using GC-MS. As no fresh living plant of *H. odoratissimum* was available, it was impossible to carry out the analysis on its spontaneous emission.

The hydrodistillation was used to extract the essential oils from the dried aerial parts (50 ± 0.1 g) of each species using a Clevenger apparatus for 2 h at 100 °C, according to the method reported in the European Pharmacopoeia [48]. The obtained oils were maintained in freezer at 4 °C and far from light sources until their analyses.

#### 3.2.2. GC-MS Analysis

Gas chromatography-electron ionization mass spectrometry (GC-EIMS) analyses were performed with a Varian CP-3800 apparatus equipped with a DB-5 capillary column (30m × 0.25 mm i.d., film thickness 0.25 μm) and a Varian Saturn 2000 ion-trap mass detector. Analytical conditions were as follows: the oven temperature was programmed rising from 60 °C to 240 °C at 3 °C/min; injector temperature 220 °C; transfer-line temperature 240 °C; carrier gas helium He (at 1 mL/min); injection of 1 μL (5% HPLC grade *n*-hexane solution); and split ratio 1:30. The acquisition parameters were as follows: full scan; scan range: 35–400 amu; and scan time: 1.0 s. Identification of the constituents was based on a comparison of (i) their retention times (t_R_) with those of the authentic samples and (ii) their linear retention indices (LRIs), determined relative to the t_R_ of the series of n-alkanes, and mass spectra with those listed in the commercial libraries (NIST 14 and ADAMS 2007) and laboratory-developed mass spectra library built up from pure substances and components of known oils and MS literature data [49,50,51,52,53,54].

### 3.3. Antimicrobial Investigation

#### 3.3.1. Antimycotic Activity

Antimycotic activity testing was carried out both on environmental potentially toxigenic molds and on pathogenic clinical isolates. In detail, isolates of *Aspergillus niger*, *Aspergillus flavus,* and *Fusarium solani,* respectively, were used. The molds were cultured from animal feedstuff.

For dermatophyte testing, clinical isolates of *Microsporum canis* and *Trichophyton mentagrophytes,* respectively, cultured from cats affected by ringworm, were employed.

All the fungal isolates were maintained onto malt extract agar (MEA) at 25 °C and identification was accomplished on the basis of their macro and microscopical morphological features.

The antimycotic activity of the selected EOs was evaluated by a microdilution test, carried out as recommended by the Clinical and Laboratory Standards Institute (CLSI) M38-A2 for molds (2008) [CLSI Reference Method for Broth Dilution Antifungal Susceptibility Testing of Filamentous Fungi. 2nd ed. CLSI; Wayne, PA, USA: 2008. approved 541 standards. CLSI document M38-A2.], with slight modification, starting from a 5% dilution. Five percent, 2.5%, 2%, 1.5%, 1%, 0.5%, 0.25%, and 0.1% dilutions in semisolid medium were achieved. All the assays were performed in triplicate. Positive and negative controls (with conventional drugs such as itraconazole and amphotericin B and with the medium without EOs) were achieved.

#### 3.3.2. Antibacterial Activity

The EOs were assayed against two clinical strains, one of *Staphylococcus aureus* and one of *Staphylococcus pseudointermedius*. Both isolates have been previously cultured from canine cutaneous specimens, typed, and stored at −80 °C in glycerol broth.

The antibacterial activity of the selected EOs, diluted at 10% in dimethyl sulfoxide (DMSO, Oxoid LTD Basingstoke, Hampshire, England), was tested by the Kirby-Bauer agar disc diffusion method [55]. A commercial disk impregnated with chloramphenicol (30 µg) (Oxoid) and a paper disk impregnated with 10 µl of DMSO were included as positive and negative controls, respectively. All tests were performed in triplicate.

The minimum inhibitory concentration (MIC) was determined for all EOs; for this purpose, the broth microdilution method was executed following the guidelines described by the Clinical and Laboratory Standard Institute [56], with some modifications as previously reported [57].

### 3.4. Statistical Analysis

The multivariate statistical analyses were carried out with the Past 3 software package’ ver. 3.14. The statistical analysis was done on each VOC and EO composition. The principal component analysis (PCA) was performed selecting the two highest principal components (PCs) of a variance/covariance matrix, methods aimed at reducing the dimensionality of the multivariate data of the matrices, while preserving most of the variance [58]. The hierarchical cluster analysis (HCA) was performed using paired group (UPGMA) algorithm and Bray-Curtis as a similarity index for both the headspace analyses and the EO contents. It is a method in which samples are considered as lying in an n-dimensional space and distances between samples are calculated, joining the object with an agglomerative procedure.

The obtained results were subjected to multivariate statistical analysis using the Past 3.14 software. Hierarchical cluster analysis (HCA) was performed with UPGMA paired group algorithm and Euclidean similarity index, while a variance and covariance matrix was used for principal component analysis (PCA).

## 4. Conclusions

The aim of this study was mainly to introduce these plants in our environment for ornamental purposes owing to the showy habitus of their architectures; moreover, as they were rich in essential oils with a pleasant odour, they may be very appreciated for their possible industrial and cosmetic applications. The chemical composition of five South African *Helichrysum* species, of which four were studied here for the first time, along with their antimicrobial activity, were reported on. The essential oils of *H. edwardsii* and *H. pandurifolium* displayed a noteworthy activity on the selected dermatophyte species. A good antibacterial action was also observed using *H. pandurifolium* and *H. trilineatum*. These EOs can be considered as good candidates for further investigations on their single compounds, seeing that the EO biological activity is strictly related to its chemical composition and to the possible synergistic effects among its components.

## Figures and Tables

**Figure 1 molecules-25-03196-f001:**
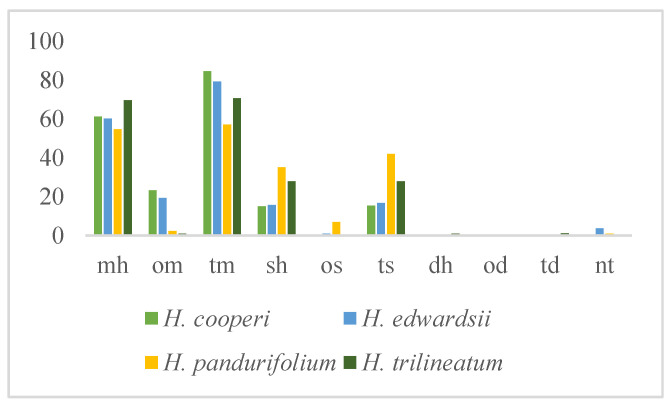
Variation of the class of compounds in the volatile organic compounds (VOCs) of the studied *Helichrysum* spp. mh: monoterpenes hydrocarbons; om: oxygenated monoterpenes; tm: total monoterpenes; sh: sesquiterpene hydrocarbons; os: oxygenated sesquiterpenes; ts: total sesquiterpenes; dh: diterpene hydrocarbons; od: oxygenated diterpenes; td: total diterpenes; nt: non-terpene derivatives.

**Figure 2 molecules-25-03196-f002:**
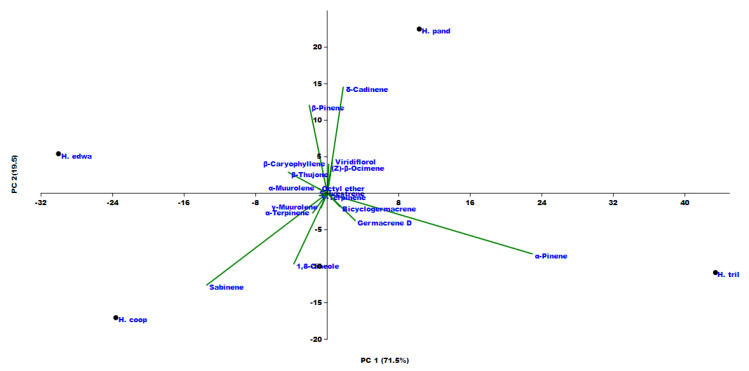
Principal component analysis (PCA) of the VOCs emitted from the investigated *Helichrysum* species.

**Figure 3 molecules-25-03196-f003:**
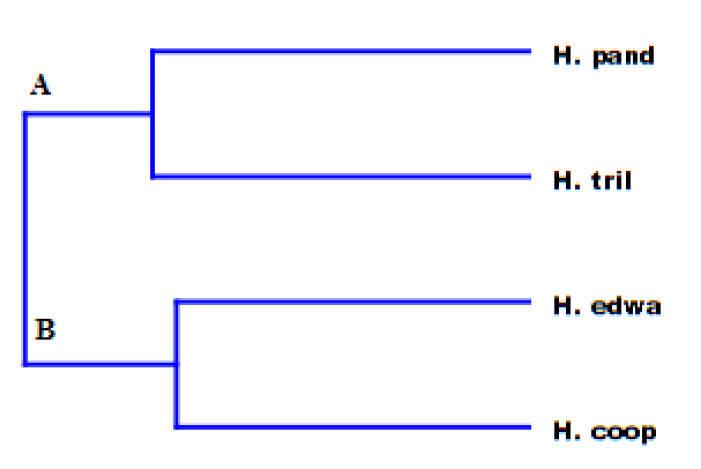
Hierarchical cluster analysis (HCA) of the VOCs emitted from the investigated Helichrysum species.

**Figure 4 molecules-25-03196-f004:**
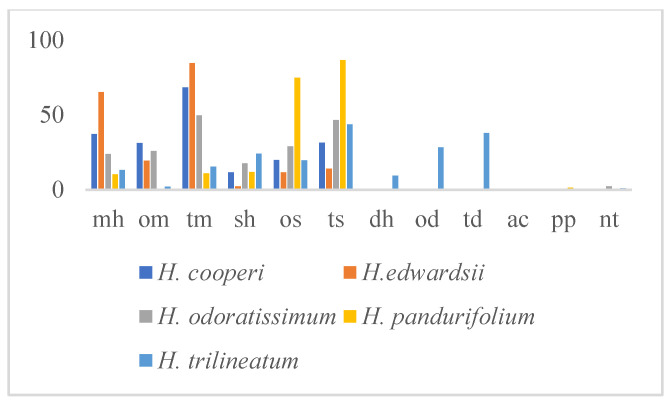
Variation of the class of compounds in the essential oils (EOs) of the studied *Helichrysum* spp. mh: monoterpenes hydrocarbons; om: oxygenated monoterpenes; tm: total monoterpenes; sh: sesquiterpene hydrocarbons; os: oxygenated sesquiterpenes; ts: total sesquiterpenes; dh: diterpene hydrocarbons; od: oxygenated diterpenes; td total diterpenes; nt: non-terpene derivatives.

**Figure 5 molecules-25-03196-f005:**
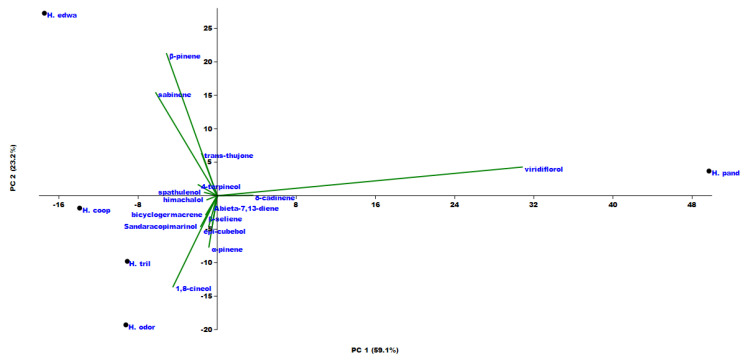
Principal component analysis of the EOs from the studied species of *Helichrysum*.

**Figure 6 molecules-25-03196-f006:**
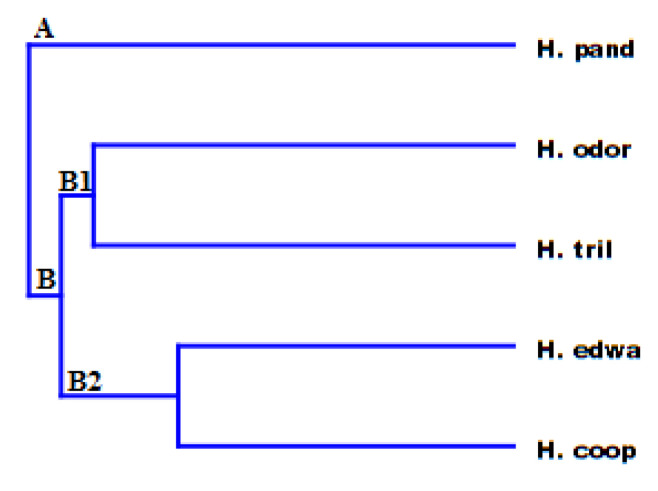
Hierarchical cluster analysis of the EOs obtained from all the studied species of *Helichrysum*.

**Table 1 molecules-25-03196-t001:** Identified compounds in spontaneous emissions of the studied *Helichysum* spp. (for chromatograms, see Appendix A).

	Compounds *	Class	RIexp	*R Lit*	*H. cooperi*	*H. edwardsii*	*H. pandurifolium*	*H. trilineatum*
					Relative Abundance (%)			
**1**	*α*-thujene	mh	930	924	2.4 ± 0.4	2.9 ± 0.1	-	0.4 ± 0.1
**2**	*α*-pinene ^∠^	mh	939	932	9.2 ± 1.0	-	**25.7 ± 4.8**	**64.8 ± 2.9**
**3**	sabinene ^∠^	mh	974	969	**34.4 ± 4.3**	**29.2 ± 1.4**	0.1 ± 0.0	0.2 ± 0.0
**4**	*β*-pinene ^∠^	mh	979	974	2.5 ± 0.2	**16.4 ± 1.7**	**17.5 ± 4.8**	2.8 ± 1.1
**5**	myrcene ^∠^	mh	991	988	-	1.1 ± 0.0	2.3 ± 1.1	0.2 ± 0.1
**6**	*α*-phellandrene ^∠^	mh	1003	1002	2.3 ± 0.5	0.5 ± 0.0	-	-
**7**	*α*-terpinene ^∠^	mh	1017	1014	6.1 ± 1.1	2.1 ± 0.2	0.1 ± 0.1	0.1 ± 0.1
**8**	*p*-cymene ^∠^	mh	1025	1020	1.5 ± 0.0	-	-	-
**9**	sylvestrene	mh	1027	1025	-	3.8 ± 0.2	-	-
**10**	limonene ^∠^	mh	1029	1024	-	-	2.0 ± 0.92	0.5 ± 0.3
**11**	1,8-cineole ^∠^	om	1031	1026	**20.5 ± 0.7**	-	1.0 ± 0.16	0.2 ± 0.1
**12**	(*Z*)-*β*-ocimene	mh	1037	1032	-	-	6.2 ± 2.84	0.1 ± 0.1
**13**	*γ*-terpinene ^∠^	mh	1060	1054	1.5 ± 0.0	3.1 ± 0.3	0.1 ± 0.06	0.4 ± 0.2
**14**	terpinolene ^∠^	mh	1088	1086	-	1.1 ± 0.1	-	-
**15**	*cis*-thujone ^∠^	om	1103	1101	-	1.3 ± 0.2	-	-
**16**	*trans*-thujone ^∠^	om	1114	1112	-	**18.0 ± 1.4**	-	-
**17**	*allo*-ocimene	om	1132	1128	-	-	1.1 ± 0.6	-
**18**	*δ*-elemene	sh	1338	1335	0.3 ± 0.1	-	-	2.4 ± 0.5
**19**	*α*-copaene ^∠^	sh	1377	1374	2.5 ± 0.3	2.1 ± 0.1	0.9 ± 0.3	0.2 ± 0.0
**20**	italicene	sh	1403	1405	1.0 ± 0.3	-	-	-
**21**	*β*-caryophyllene ^∠^	sh	1419	1417	0.7 ± 0.0	6.9 ± 0.8	6.8 ± 1.9	4.0 ± 0.1
**22**	*cis*-thujopsene	sh	1431	1429	-	-	-	-
**23**	*β*-copaene	sh	1432	1430	0.3 ± 0.0	-	-	1.0 ± 0.1
**24**	*α*-humulene ^∠^	sh	1455	1452	0.3 ± 0.0		0.3 ± 0.2	1.9 ± 0.2
**25**	*allo*-aromadendrene	sh	1460	1458	-	0.4 ± 0.0	0.9 ± 0.3	0.3 ± 0.0
**26**	*γ*-muurolene	sh	1480	1478	4.1 ± 0.5	-	0.2 ± 0.1	0.2 ± 0.0
**27**	germacrene D	sh	1485	1484	1.4 ± 0.1	-	0.1 ± 0.1	**9.7 ± 3.6**
**28**	*α*-muurolene	sh	1499	1500	-	4.4 ± 0.2	-	-
**29**	bicyclogermacrene	sh	1500	1500	0.8 ± 0.2	-	-	4.5 ± 1.5
**30**	*δ*-cadinene	sh	1524	1522	-	0.4 ± 0.0	**22.0 ± 4.5**	0.2 ± 0.0
**31**	viridiflorol	os	1591	1592	-	-	6.9 ± 1.6	-
**32**	octyl ether	nt	1659	1657 ^§^	-	3.7 ± 0.0	-	-

* Compounds present with percentage ≥1% in at least one of the *Helichrysum* spp. Data are reported as mean values (*n* = 3 ± SD); L.R.I^exp^, linear retention time experimentally determined; L.R.I.^lit^, linear retention time reported by Adams 2007 (26); ^§^ linear retention time in pubchem (www.pubchem.ncbi.nlm.nih.gov). ^∠^ compounds identified by comparison to injected authentic reference samples purchased from Sigma-Aldrich, Inc. Bold format: Main constituents.

**Table 2 molecules-25-03196-t002:** Class of compounds in spontaneous emissions of the studied *Helichysum* spp.

Chemical Classes	*H. cooperi*	*H. edwardsii*	*H. pandurifolium*	*H. trilineatum*
	Relative Abundance (%)			
Monoterpene Hydrocarbons (mh)	61.2 ± 2.7	60 ± 2.5	54.7 ± 4.6	69.6 ± 4.7
Oxygenated Monoterpenes (om)	23.3 ± 0.7	19.3 ± 1.5	2.3 ± 0.83	1.0 ± 0.1
**Total Monoterpenes (TM)**	**84.5**	**79.3**	**57.0**	**70.6**
Sesquiterpene Hydrocarbons (sh)	15.1 ± 1.7	15.8 ± 1.0	35.2 ± 3.9	27.9 ± 4.7
Oxygenated Sesquiterpenes (os)	0.2 ± 0.0	1.0 ± 0.0	6.9 ± 1.6	-
**Total Sesquiterpenes (TS)**	**15.3**	**16.8**	**42.1**	**27.9**
Diterpene Hydrocarbons (dh)	-	-	-	1.0 ± 0.2
Oxygenated Diterpenes (od)	-	-	-	0.1 ± 0.1
**Total Diterpenes (TD)**	-	-	-	**1.1**
**Non-terpene Derivatives (nt)**	**0.2 ± 0.2**	**3.7 ± 0.0**	**0.9 ± 0.1**	**0.4 ± 0.2**
**Total Identified (%)**	**100.0 ± 0.0**	**99.8 ± 0.2**	**100.0 ± 0.0**	**100.0 ± 0.0**

**Table 3 molecules-25-03196-t003:** Identified compounds in the essential oils of the studied *Helichysum* spp. (for chromatograms, see Appendix A).

	Compounds *	Class	LRI^exp^	LRI^lit^	*H. cooperi*	*H. edwardsii*	*H. odoratissimum*	*H. pandurifolium*	*H. trilineatum*
					Relative Abundance (%)				
**1**	*α*-thujene	mh	930	924	5.0 ± 0.4	0.6 ± 0.0	-	-	-
**2**	*α*-pinene ^∠^	mh	939	939	2.3 ± 0.1	4.6 ± 0.1	**16.5 ± 0.7**	5.8 ± 0.0	**11.5 ± 2.9**
**3**	sabinene ^∠^	mh	975	969	**14.7 ± 1.2**	**22.4 ± 0.1**	0.2 ± 0.1	-	-
**4**	*β*-pinene ^∠^	mh	979	974	4.3 ± 0.1	**31.2 ± 3.1**	2.0 ± 0.0	2.9 ± 0.1	1.5 ± 0.2
**5**	myrcene ^∠^	mh	991	988	0.5 ± 0.0	0.7 ± 0.0	0.4 ± 0.2	0.2 ± 0.0	-
**6**	*α*-terpinene ^∠^	mh	1017	1014	3.6 ± 0.3	1.0 ± 0.1	2.2 ± 0.1	-	-
**7**	*p*-cymene ^∠^	mh	1025	1025	0.7 ± 0.2	0.4 ± 0.2	0.4 ± 0.2	-	-
**8**	limonene ^∠^	mh	1029	1024	-	2.3 ± 0.9	-	0.2 ± 0.0	0.1 ± 0.0
**9**	1,8-cineol ^∠^	om	1031	1026	**16.4 ± 0.1**	-	**25.1 ± 0.9**	0.6 ± 0.3	-
**10**	(*Z*)-*β*-ocimene	mh	1037	1032	-	-	-	1.1 ± 0.2	-
**11**	*γ*-terpinene ^∠^	mh	1060	1054	4.7 ± 0.4	1.5 ± 0.5	1.2 ± 0.0	-	0.1 ± 0.1
**12**	*cis*-sabinene hydrate ^∠^	om	1070	1065	2.6 ± 0.1	0.7 ± 0.1	-	-	-
**13**	terpinolene ^∠^	mh	1089	1086	1.2 ± 0.2	0.4 ± 0.0	0.5 ± 0.0	-	-
**14**	*trans*-sabinene hydrate	om	1098	1098	2.3 ± 0.1	0.5 ± 0.0	-	-	-
**15**	*cis*-thujone ^∠^	om	1103	1101	-	1.3 ± 0.3	-	-	-
**16**	*trans*-thujone ^∠^	om	1114	1112	-	**8.7 ± 1.2**	-	-	-
**17**	*trans*-pinocarveol	om	1139	1135	-	0.9 ± 0.1	-	-	0.6 ± 0.1
**18**	sabina ketone	om	1159	1154	-	0.5 ± 0.0	-	-	-
**19**	pinocarvone	om	1165	1160	-	0.5 ± 0.1	-	-	0.3 ± 0.0
**20**	terpinen-4-ol ^∠^	om	1177	1174	**8.4 ± 0.9**	3.4 ± 1.0	0.7 ± 0.0	-	0.2 ± 0.0
**21**	myrtenol	om	1195	1194	0.4 ± 0.0	1.4 ± 0.4	-	-	
**22**	*β*-bourbonene	sh	1388	1387	0.3 ± 0.1	-	-	-	0.5 ± 0.1
**23**	*β*-caryophyllene ^∠^	sh	1419	1417	0.9 ± 0.5	1.8 ± 0.0	4.1 ± 0.1	1.2 ± 0.1	0.4 ± 0.1
**24**	*γ*-elemene	sh	1433	1434	-	-	-	-	0.7 ± 0.6
**25**	aromadendrene ^∠^	sh	1441	1439	0.2 ± 0.0	-	-	-	0.9 ± 0.1
**26**	*γ*-muurolene	sh	1480	1478	0.5 ± 0.0	-	1.8 ± 0.5	-	0.8 ± 0.1
**27**	*ar*-curcumene	sh	1481	1479	1.2 ± 0.0	-	-	-	
**28**	germacrene D	sh	1485	1484	3.7 ± 0.4	-	-	-	5.5 ± 0.4
**29**	aristolochene	sh	1487	1487	-	-	-	0.5 ± 0.0	
**30**	*β*-selienene	sh	1490	1489	-	-	**8.1 ± 0.1**	0.7 ± 0.3	0.7 ± 0.2
**31**	*epi*-cubebol	os	1494	1493	-	-	**9.0 ± 0.1**		
**32**	viridiflorene	sh	1497	1496	-	-	1.7 ± 0.2	0.6 ± 0.1	
**33**	bicyclogermacrene	sh	1500	1500	1.0 ± 0.1	-	-	-	**10.7 ± 0.2**
**34**	*γ*-cadinene	sh	1513	1513	-	-	-	-	0.5 ± 0.1
**35**	*trans*-*γ*-cadinene	sh	1514	1513 ^$^	0.6 ± 0.1	-	0.1 ± 0.0	0.1 ± 0.1	-
**36**	*δ*-cadinene	sh	1524	1522	0.8 ± 0.0	-	0.7 ± 0.1	7.5 ± 0.8	1.1 ± 0.1
**37**	*cis*-sesquisabinene hydrate	os	1544	1542	0.6 ± 0.1	-	-	-	-
**38**	palustrol	os	1568	1567	-	-	-	0.2 ± 0.0	3.4 ± 3.0
**39**	germacrene d-4-ol	os	1574	1574	-	-	-	2.5 ± 1.2	-
**40**	spathulenol	os	1578	1577	1.1 ± 0.2	3.3 ± 0.3	-	0.3 ± 0.1	7.2 ± 0.6
**41**	caryophyllene oxide ^∠^	os	1583	1582	1.6 ± 0.3	4.6 ± 0.6	-	0.6 ± 0.3	-
**42**	globulol	os	1585	1590	2.3 ± 0.5	-	1.1 ± 0.00	-	1.6 ± 0.1
**43**	viridiflorol	os	1593	1592	-	-	2.9 ± 0.14	**60.3 ± 0.8**	0.5 ± 0.1
**44**	octadienyl tiglate, 2*E*, 4*E*-	nt	1595	1595 ^!^	-	-	2.1 ± 0.03	-	-
**45**	rosifoliol	os	1600	1600	-	-	-	0.6 ± 0.0	0.7 ± 0.3
**46**	1-*epi*-cubenol	os	1629	1627	-	-	0.8 ± 0.01	-	-
**47**	muurola-4,10(14)-dien-1*β*-ol	os	1635	1630	-	-	-	0.8 ± 0.0	0.5 ± 0.0
**48**	*β*-acorenol	os	1636	1632	-	-	0.5 ± 0.01	-	-
**49**	isospathulenol	os	1638	1639 ^$^	-	-	-	-	0.9 ± 0.1
**50**	*epi*-*α*-cadinol	os	1640	1638	-	-	-	-	0.8 ± 0.1
**51**	caryophylla-4(14),8(15)-dien-5-ol	os	1641	1639 ^$^	-	0.9 ± 0.0	-	-	-
**52**	*epi*-*α*-muurolol	os	1642	1640	0.1 ± 0.1	-	-	0.5 ± 0.1	-
**53**	valerianol	os	1644	1656	-	0.5 ± 0.0	-	-	-
**54**	*β*-eudesmol ^∠^	os	1651	1649	-	1.0 ± 0.3	-	-	-
**55**	*α*-Cadinol	os	1653	1652	-	-	-	-	0.9 ± 0.0
**56**	pogostol	os	1655	1651	-	-		4.8 ± 1.8	-
**57**	himachalol	os	1657	1652	6.6 ± 0.5	-	0.2 ± 0.0	-	0.7 ± 0.5
**58**	selin-11-en-4*α*-ol	os	1660	1658	-	-	0.6 ± 0.3	-	-
**59**	intermedeol	os	1667	1665	3.0 ± 0.2	-	-	-	-
**60**	14-hydroxy-9-*epi*- (*E*)-caryophyllene	os	1670	1668	-	-	5.0 ± 0.3	-	-
**61**	valeranone	os	1672	1674	-	1.2 ± 0.2	-	-	-
**62**	*β*-bisabolol	os	1675	1674	0.8 ± 0.0	-	-	-	-
**63**	aromadendrene oxide-(2)	os	1678	1678 ^$^	-	-	-	2.9 ± 0.5	0.3 ± 0.0
**64**	*epi*-*α*-bisabolol	os	1685	1683	2.6 ± 0.6	-	5.6 ± 0.1	-	-
**65**	*ent*-germacra-4(15),5,10(14)-trien-1*β*-ol	os	1695	1696 ^$^	-	-	-	0.8 ± 0.1	0.5 ± 0.1
**66**	14-hydroxy-*α*-humulene	os	1713	1713	-	-	1.4 ± 0.4	-	-
**67**	2,4b-dimethyl-8-methylene-2-vinyl-1,2,3,4,4a,4b,5,6,7,8,8a,9-dodecahydrophenanthrene	nt	1842	1858 *	-	-	-	-	0.6 ± 0.2
**68**	beyerene	dh	1943	1951 ^$^	-	-	-	-	1.0 ± 0.2
**69**	(*E*)-1(6,10-dimethylundec-5-en-2-yl)-4-methylbenzene	pp	1951	1950 ^$^	-	-	-	1.4 ± 0.1	-
**70**	13-*epi*-manoyl oxide	od	2011	2002 ^$^	-	-	-	-	0.6 ± 0.1
**71**	*cis*-3,14-clerodadien-13-ol	od	2051	2051 ^$^	-	-	-	-	1.2 ± 0.3
**72**	abietatriene	dh	2054	2055	-	-	-	-	1.0 ± 0.4
**73**	abieta-7,13-diene	dh	2080	2088 ^$^	-	-	-	-	6.1 ± 0.9
**74**	sandaracopimarinal	od	2214	2184	-	-	-	-	1.2 ± 0.2
**75**	sclareol ^∠^	od	2227	2222	-	-	-	-	0.6 ± 0.1
**76**	pimara-7,15-dien-3-ol	od	2253	2253 ^$^	-	-	-	-	0.6 ± 0.1
**77**	larixol	od	2264	2265	-	-	-	-	0.6 ± 0.1
**78**	sandaracopimarinol	od	2279	2269	-	-	-	-	**17.7 ± 2.3**
**79**	neoabietal	od	2320	2299 ^$^	-	-	-	-	1.3 ± 0.2
**80**	neoabietic acid	od	2335	2335 ^$^	-	-	-	-	2.1 ± 0.4
**81**	abietinol	od	2391	2389 ^$^	-	-	-	-	1.2 ± 0.2
	**N° of Identified Peaks**				**32**	**26**	**27**	**24**	**43**
	**EO Yield (%*w*/*w*)**				**0.60 ± 0.0**	**0.57 ± 0.4**	**0.45 ± 0.0**	**0.48 ± 0.1**	**0.1 ± 0.0**

* Compounds present with percentage ≥1% in at least one of the *Helichrysum* spp. Data are reported as mean values (*n* = 3 ± SD); EO, essential oil; L.R.I.^exp^, linear retention time experimentally determined; L.R.I.^lit^, linear retention time reported by Adams 2007; ^$^ linear retention time reported by NIST 2014; * linear retention index in chemspider (www.chemspider.com); ^!^ linear retention time reported by Adams and Dev 2010 [23]. ^∠^ compounds identified by comparison to injected authentic reference samples purchased from Sigma-Aldrich, Inc. Bold format: Main constituents.

**Table 4 molecules-25-03196-t004:** Class of compounds in the essential oil of the studied *Helichysum* spp.

Chemical Classes	*H. cooperi*	*H. edwardsii*	*H. odoratissimum*	*H. pandurifolium*	*H. trilineatum*
	Relative Abundance (%)				
Monoterpene Hydrocarbons (mh)	37.1 ± 1.2	65.1 ± 2.4	23.7 ± 1.0	10.3 ± 0.4	13.2 ± 3.4
Oxygenated Monoterpenes (om)	31.1 ± 0.8	19.4 ± 0.6	25.8 ± 2.0	0.6 ± 0.4	2.1 ± 0.1
**Total Monoterpenes (TM)**	**68.2**	**84.5**	**49.5**	**10.9**	**15.3**
Sesquiterpene Hydrocarbons (sh)	11.6 ± 0.9	2.3 ± 0.2	17.5 ± 0.3	11.8 ± 0.4	24.1 ± 0.9
Oxygenated Sesquiterpenes (os)	19.8 ± 0.5	11.6 ± 1.0	28.9 ± 1.6	74.7 ± 0.8	19.5 ± 3.5
**Total Sesquiterpenes**	**31.4**	**13.9**	**46.4**	**86.5**	**43.6**
Diterpene Hydrocarbons (dh)	-	-	-	-	9.5 ± 1.6
Oxygenated Diterpenes (od)	-	0.5 ± 0.0	-	0.4 ± 0.1	28.2 ± 3.2
**Total Diterpenes**	-	**0.5**	-	**0.4**	**37.7**
Apocarotenoids (ac)	-	-	-	-	0.2 ± 0.1
Phenylpropanoids (pp)	-	-	-	1.4 ± 0.1	-
Non-terpene Derivatives (nt)	-	-	2.1 ± 0.0	0.3 ± 0.1	0.9 ± 0.2
**Total Identified (%)**	**99.6 ± 0.4**	**98.9 ± 1.1**	**98.0 ± 2.0**	**99.5 ± 0.5**	**97.7 ± 2.3**

**Table 5 molecules-25-03196-t005:** Antifungal activity (in %) of the tested essential oils (EOs) of *Helichrysum* species.

		Dermatophytes		*Aspergillus*		*Fusarium*
		*M. canis*	*T. mentagrophytes*	*A. flavus*	*A. niger*	*F. solani*
**Present study**	***H. cooperi***	0.5	1	>5	>5	>5
	***H. edwardsii***	0.25	1	>5	>5	>5
	***H. odoratissimum***	>5	>5	>5	>5	>5
	***H. pandurifolium***	0.25	0.5	5	>5	5
	***H. trilineatum***	5	5	5	>5	5
**Previous study ***	***H. foetidum***	>5	>5	>5	>5	>5
	***H. patulum***	5	5	>5	>5	>5

* Reference list number [22].

**Table 6 molecules-25-03196-t006:** Antibacterial activity of the tested EOs of *Helichrysum* species. MIC, minimum inhibitory concentration.

		*S. aureus*		*S. pseudointermedius*	
		*Disk (mm)*	*MIC (%)*	*Disk (mm)*	*MIC (%)*
**Present study**	***H. cooperi***	0	>10	0	>10
	***H. edwardsii***	0	>10	7	10
	***H. odoratissimum***	0	>10	0	>10
	***H. pandurifolium***	7	5	8	5
	***H. trilineatum***	7	5	8	5
**Previous study ***	***H. foetidum***	0	>10	0	>10
	***H. patulum***	0	>10	0	>10

* Reference list number [22].

**Table 7 molecules-25-03196-t007:** Botanical characteristics and traditional medicinal use of the investigated species of *Helichrysum*.

Studied Species	Photo	Botanical Characteristics and Folk Uses
***H. cooperi* Harv.**	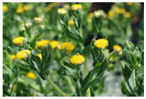	**Voucher N°: HMGBH.e/9006.2019.002** **General information** It prefers the prairies and the woods border Green, simple, oblong leaves with entire margin and evident vein, covered by a fine sticky fuzz. The fresh leaves have a strong aromatic smell. The flower heads are golden yellow placed in a large inflorescence. The fruit is a cypsela **Use in traditional African medicine** Leaves are used as a fumigant and as part of snake bite remedy [2].
***H. edwardsii* Wild**	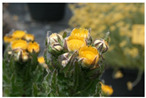	**Voucher N°: HMGBH.e/9006.2017.002** **General information** It is a high suffrutice plant, which reaches more than 1 m higher Robust, woody stems. The leaves are oblong or ovate, with the surface covered with glandular hairs. Globular flower heads are gathered in a corymb inflorescence. The fruit are achenes of about 1 mm **Use in traditional African medicine** Leaves are used as a fumigant and as part of snake bite remedy
***H. odoratissimum* (L.) Sweet**	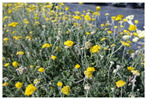	**Voucher N°: HMGBH.e/9006.2019.007** **General information** It is widespread in the highlands, mountains, and coastal areas, particularly on grassy and rocky slopes. The stems are thin and fluffy. Grey-white leaves can be oblong, lanceolate, ligulated, or spatulate, with generally obtuse apex. The flower heads are yellow, small, numerous, and gathered in terminal corymbs carried by a bare peduncle. The fruits are opaque brown. Flowering occurs mainly from August to December in the south-west of the Cape Province and from January to June in the other South African regions **Use in traditional African medicine** Leaf infusions and decoctions are used to treat fever, insomnia, colic, menstrual pain, and sterility in women; root extracts are used as purgatives; while leaf extracts are used as eye drops to treat conjunctivitis [2].
***H. pandurifolium* Schrank (= *H. auriculatum* Less.)**	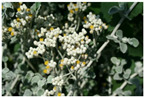	**Voucher N°: HMGBH.e/9006.2018.004** **General information** Grows in sandy or rocky areas, from sea level up to 1500 m of altitude. The stems are long, thin, slightly grey, and with a woolly pubescence. Orbicular or ovate grey and woolly leaves have a sinuous margin, and shrink abruptly to form a narrow and curved base similar to a petiole. The yellow flower heads are gathered in terminal corymbs. The involucre bracts are imbricate, sometimes light brown outside; the inner ones are white-pink and with a sharp apex and exceed the flowers in length. The fruits are achenes, which carry a pappus with many feathery bristles. Flowering occurs between September and January **Use in traditional African medicine** Infusions and ointments obtained from the aerial parts are used to treat respiratory and heart problems, back pain, and kidney stones [2].
***H. trilineatum* DC.**	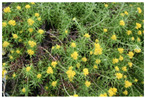	**Voucher N°: HMGBH.e/9006.2019.006** **General information** Rocky outcrops of the steep mountain slopes and on the highlands. Tomentose and grey-white stems. Sessile, linear or oblong leaves, with mucronate and curved apex and revolute margins. The upper face is dotted with secretory glands, the lower face is white and woolly and crossed by parallel veins. The yellow flower heads, gathered in terminal corymbs. Flowering occurs between August and February. The involucre bracts are imbricate, of golden brown colour on the outside and intense yellow on the inside. The fruits are achenes of 1 mm long ending in a silky pappus. **Use in traditional African medicine** chloroform extract is active against *Staphylococcus aureus* [46].

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
